# Impact of Dietary Patterns on *H. pylori* Infection and the Modulation of Microbiota to Counteract Its Effect. A Narrative Review

**DOI:** 10.3390/pathogens10070875

**Published:** 2021-07-10

**Authors:** Ascensión Rueda-Robles, Teresa Rubio-Tomás, Julio Plaza-Diaz, Ana I. Álvarez-Mercado

**Affiliations:** 1Center of Biomedical Research, Institute of Nutrition and Food Technology “José Mataix”, University of Granada, Avda. del Conocimiento s/n., Armilla, 18016 Granada, Spain; ruedarobles@ugr.es; 2Institut d’Investigacions Biomèdiques August Pi i Sunyer (IDIBAPS), 08036 Barcelona, Spain; teresa.rubio.t111@gmail.com; 3School of Medicine, University of Crete, 70013 Heraklion, Crete, Greece; 4Children’s Hospital Eastern Ontario Research Institute, Ottawa, ON K1H 8L1, Canada; jrplaza@ugr.es; 5Department of Biochemistry and Molecular Biology II, School of Pharmacy, University of Granada, 18071 Granada, Spain; 6Instituto de Investigación Biosanitaria ibs.GRANADA, Complejo Hospitalario Universitario de Granada, 18014 Granada, Spain

**Keywords:** *Helicobacter pylori*, diet, gastrointestinal microbiota, experimental models, therapeutic approaches

## Abstract

*Helicobacter pylori (H. pylori)* is a Gram-negative bacterium that colonizes the stomach and can induce gastric disease and intra-gastric lesions, including chronic gastritis, peptic ulcers, gastric adenocarcinoma, and mucosa-associated lymphoid tissue lymphoma. This bacterium is responsible for long-term complications of gastric disease. The conjunction of host genetics, immune response, bacterial virulence expression, diet, micronutrient availability, and microbiome structure influence the disease outcomes related to chronic *H. pylori* infection. In this regard, the consumption of unhealthy and unbalanced diets can induce microbial dysbiosis, which infection with *H. pylori* may contribute to. However, to date, clinical trials have reported controversial results and current knowledge in this field is inconclusive. Here, we review preclinical studies concerning the changes produced in the microbiota that may be related to *H. pylori* infection, as well as the involvement of diet. We summarize and discuss the last approaches based on the modulation of the microbiota to improve the negative impact of *H. pylori* infection and their potential translation from bench to bedside.

## 1. Introduction

*Helicobacter pylori*, considered the main bacterium responsible for gastric disease and long-term complications [1], is a Gram-negative bacterium that colonizes the stomach [2]. Infection with *H. pylori* is very prevalent around the world. It is estimated that 48.6% of the global adult population is affected [3]. All patients infected with *H. pylori* suffer from inflammation of the gastric mucosa (chronic gastritis), while, in some cases, the disease becomes more developed (e.g., peptic ulcer, gastric adenocarcinoma, and mucosa-associated lymphoid tissue lymphoma) [2,4]. In fact, *H. pylori* is responsible for 90–95% of duodenal ulcers and 70–85% of gastric ulcers [5]. 

The most important virulence factors of *H. pylori* are cytotoxin-associated A (CagA) and vacuolating cytotoxin (VacA). The expression of these proteins varies among *H. pylori* strains [6]. In the strains that carry the *CagA* gene, the CagA protein enters the host gastric cells upon *H. pylori* secretion, where it is phosphorylated and causes changes in cell morphology that lead to disturbed cell function by affecting multiple signaling pathways. CagA is related to higher rates of peptic ulcer disease and a higher probability of developing a more aggressive inflammatory disease and gastric carcinoma [7]. In the strains that express the *VacA* gene, *H. pylori* secretes the pore-forming protein VacA, which enters the host cells by endocytosis [6]. VacA accumulation in different host cell compartments can have many consequences related to gastric inflammation and, therefore, gastric carcinogenesis, such as the induction of apoptosis [8] and autophagic cell death [9], the disruption of gastric epithelial cell tight junctions [10], the suppression of host T cell proliferation and activation by yet to be discovered mechanisms [11], and the modulation of host cell metabolism [12]. 

*H. pylori* produces ammonia through urease activity, protecting itself from gastric acidity and other enzymes, such as phospholipase A2 and C, and glucosulphatase, which handle gastric mucosal damage [13] and chronic superficial gastritis, leading to atrophy of the gastric glands and resulting in reduced gastric acid secretion. Infection with *H. pylori* is also associated with anemia, as it impairs iron absorption as a result of chronic gastritis and gastric hypochlorhydria [3]. Infection during early childhood is associated with an augmented risk of coeliac disease [14]. Findings from epidemiologic studies point to *H. pylori* as an initiator of gastric cancer [2,15]. Moreover, this bacterium has been included as a class I carcinogen of gastric cancer development in the World Health Organization’s International Agency for Research on Cancer since 1994 [16]. The effect of *H. pylori* on oncogenesis is described by two mechanisms: (1) an indirect inflammatory reaction to *H. pylori* infection on the gastric mucosa, and (2) a direct epigenetic outcome of *H. pylori* on gastric epithelial cells development [17]. Consequently, *H. pylori* infection impairs the gastric tissue microenvironment, and promotes epithelial-mesenchymal transition and further gastric cancer progression [17]. In this line, dietary habits such as a high intake of green tea, fruits, or vegetables showed benefits against gastric cancer risk. Moreover, carotenoids, folate, vitamin C, and phytochemicals from fruits and vegetables seem to exert a protective role in carcinogenesis. On the contrary, salt and the availability of some transition metals can alter *H. pylori* virulence and accelerate carcinogenesis [18]. In addition, in the context of chronic *H. pylori* infection, the intersection of host genetics, immune response, bacterial virulence expression, diet, micronutrient availability, and microbiome structure influence disease outcomes [18] (Figure 1).

With the aforementioned in mind, one might suggest that changes in diet patterns, aimed to modulate microbiota in *H. pylori* carriers, might be exploited to improve disease risk and promote gastric health. However, the relationship of these variables remains poorly understood. In the present review, we aimed to shed light on current knowledge about the changes in microbiota induced by *H. pylori* infection and the impact of diet, as well as to summarize the recent approaches aimed at modulating the microbiota to avoid and reduce the negative effects caused by *H. pylori*.

## 2. The Impact of Diet Patterns in *Helicobacter pylori* Infection

After infection, *H. pylori* can induce malabsorption of several nutrients [19] and affect the physiological regulation of the intestinal metabolic hormones, such as ghrelin and leptin, which are involved in food intake, energy expenditure, and body mass [20]. Low circulating ghrelin levels were found in *H. pylori*-positive patients and in those with the more virulent *H. pylori* strain [21,22]. Infected individuals can also suffer irreversible inactivation of vitamin C through the effect of hypochlorhydria due to gastric atrophy, as intragastric pH levels increase, turning vitamin C into its less active form [23]. 

Many substances routinely taken in via diet have potent antibacterial activity against *H. pylori* and may reduce its potential for colonization [24]. By contrast, diet could affect the development and progression of *H. pylori* infection, because it alters the gastric environment through the host’s nutrient intake. For instance, animal studies indicated that diets with high salt intake promoted *H. pylori* colonization by disturbing the integrity and viscosity of the gastric mucosa, causing an inflammatory state that led to increased epithelial damage, hypochlorhydria, and gastric cancer [18,25]. 

Animal studies showed not only an increase in colonization capable of producing losses of parietal cells, atrophy, and intestinal metaplasia [26,27], but also high levels of gastric inflammation and oxidative stress [28]. A high-salt diet could influence gastric cancer risk by modulating *H. pylori* gene expression due to transcriptional alterations. In an in vitro study conducted by Loh et al., 65 genes were upregulated and 53 genes were downregulated in response to high salinity conditions [29]. The in vitro study of Voss et al. also showed that the bacterial membrane proteome was altered in response to salinity conditions [30]. 

By contrast, the results of sulforaphane consumption (present in broccoli sprouts) in female C57BL/6 mice infected with *H. pylori* Sydney Strain 1 subjected to a high-salt diet produced a reduction in gastric bacterial colonization, attenuated mucosal expression of tumor necrosis factor-alpha and interleukin (IL)-1beta, mitigated corpus inflammation, and prevented the expression of high-salt-induced gastric corpus atrophy [31]. More recently, an in vitro study showed that the concentration of salt in the environment influenced the composition of the *H. pylori* exoproteome and led to increased levels of a secreted VacA toxin, which would increase the risk of gastric cancer associated with the consumption of a high-salt diet [32]. In a Mongolian gerbil model, infected animals fed with diets rich in salt developed gastric ulcers significantly more frequently than those consuming a normal salt diet, and lower hemoglobin levels were found in infected gerbils consuming a high-salt and low-iron diet [33] compared to controls. 

In this animal study, dietary consumption of walnuts rich in n-3 polyunsaturated fatty acids were proposed as a nutritional intervention to prevent *H. pylori* associated with gastric cancer. The results of walnut consumption, using a model of *H. pylori*-initiated gastric carcinogenesis promoted by a high-salt diet in mice, showed a significant improvement in chronic atrophic gastritis and a significant decrease in tumorigenesis compared to the control group [34]. Similarly, a mouse model of chronic atrophic gastritis and gastric tumorigenesis initiated by *H. pylori* and promoted by high salt content was used by Jeong et al. to evaluate the efficacy of long-term dietary administration of artemisia and green tea extracts, resulting in improvements in both gastritis and tumorigenesis. These authors found decreased expression of cyclooxygenase-2 (COX-2), tumor necrosis factor-α (TNF-α), IL-6, lipid peroxide, and activated signal of both the transducer and the activator of transcription 3 (STAT3) (relevant to *H. pylori* infection), as well as serum activations of TNF-α and nuclear factor kappa B [35].

Iron is an essential growth factor for *H. pylori* [36]. Several studies have associated *H. pylori* infection and iron deficiency anemia in human and animal studies [37,38]. This may be due to the fact that *H. pylori*-related gastritis leads to a defect in its absorption, or increases gastric pH and alters the reduction of the ferric to ferrous form (essential for the absorption of non-heme iron) [39,40]. Animals fed with low iron diets displayed increased infiltration of immune cells at the site of infection, more rapid onset of gastritis, and a higher rate of cancer development [41]. Results from a meta-analysis by Kin et al., indicated that red and processed meats rich in heme iron led to endogenous production of N-nitro compounds and increased progression of *H. pylori* infection by causing DNA damage and oxidative stress (a significant risk factor for gastric cancer) [42,43]. This increase in oxidative stress can be further aggravated in the stomach of individuals affected by *H. pylori* [44]. Thus, including sulfur-rich food, such as broccoli sprouts, in the diet can not only protect cells from oxidative stress damage, but also inhibit the viability of *H. pylori* and mitigate gastritis, as indicated in this study conducted in mice and humans [31]. Similarly, it was shown that, in a mouse model of infection by *H. pylori*, resveratrol exerted a beneficial effect against *H. pylori*-associated gastritis by combating oxidative stress [45], thus the consumption of resveratrol-rich foods may also have a positive effect.

As far as the consumption of cured, pickled, and smoked foods is concerned, nitrosamines (which are often present in this type of food) exert a synergistic effect with *H. pylori* by inducing intestinal metaplasia and intraepithelial neoplasia in a model of non-human primates through changes in multiple genes associated with cancer [46]. 

It is well known that high-fat diets (HFD), high-ultra-processed foods and low intake of fruits and vegetables are key factors for the development of obesity. The risk of *H. pylori* infection is not increased in young overweight individuals [47], but high-carbohydrate and sweetened diets were positively associated with an increased prevalence [48], whereas whole-grains, roots and tubers, vegetables, mushrooms, miscellaneous beans, vegetable oils, nuts, and seeds consumed in abundance were linked to a reduced risk of infection in humans [49]. 

The rejuvenation of atrophic gastritis and prevention of tumorigenesis in mice, via cancer-preventive kimchi made from a changed recipe from the Research Institute at Pusan National University [50], was also promoted by the same authors. In sum, many promising results show the importance of the impact of diet on *H. pylori* infection, although there are many controversies in this respect that need to be elucidated.

### Diet, Helicobacter pylori, and the Gut Microbiota 

Gut microbiota (defined as the community of microorganisms that subsists within the digestive tract) plays both local and remote roles in important physiological processes, particularly inflammation and immune response [51]. Diet has a considerable effect on the composition of gastrointestinal microbiota and an unhealthy diet, among other factors, alters its composition and induces gastric microbial dysbiosis [52]. The gut microbiome is the genetic material of all the microbes that live on and inside the human body [53,54,55]. *H. pylori* colonization results in the alteration of gastric microbiota and a reduction in bacterial diversity [56,57], thus modifying the host’s gut microbiome [58]. A growing number of studies reported greater ecosystem diversity in the gastrointestinal tract and associated *H. pylori* presence with variations within the structure of the microbiome [59]. Gut microbiome changes triggered by the first *H. pylori* acquisition could also corroborate the host immunological status and, as a result, the manifestation of several diseases. This is important due to the fact that gut microbiota is modified by antibiotic treatment, which also decreases *H. pylori*-induced inflammation [60].

Some studies have evaluated changes in the microbial profile related to *H. pylori* in humans [61,62,63]. For instance, individuals infected by *H. pylori* present higher levels of *Succinivibrio*, *Coriobacteriaceae*, *Enterococcaceae*, and *Rikenellaceae*, and an increased abundance of *Candida glabrata* and other unclassified fungi [62]. In children, gastric commensal bacteria are altered during *H. pylori* infection [61]. Indeed, *H. pylori* dominated the microbial community in infected children, while *H. pylori*-negative individuals showed a relative abundance of *Gammaproteobacteria*, *Betaproteobacteria*, *Bacteroidia* and *Clostridia* classes and a higher bacterial richness and diversity [63]. However, the relationship and underlying mechanisms involved need to be elucidated. 

Concerning animal studies, a cocoa-supplemented diet was shown to modify the intestinal microbiota and health-promoting microbiota in diabetic rats [64], whereas the consumption of diets high in sugar induced gut dysbiosis [65]. Along the same line, the dietary pattern known as the Mediterranean diet is characterized by the high consumption of vegetables, fruits, whole grains, legumes, nuts, fish, lean meats, and virgin olive oil. This diet is associated with a health-promoting microbiota able to prevent the appearance of non-communicable diseases or cancer [66]. In contrast, an HFD produces an increase in bacteria expressing lipopolysaccharides in the gut microbiota, resulting in a pro-inflammatory state, weight gain, and increased intestinal permeability [67]. It is not clear whether the diet, by modifying the intestinal microbiota, creates helpful conditions for colonization by *H. pylori*, as many factors are involved. However, it is known that *H. pylori*-induced alterations in the composition of the intestinal microbiota are due to changes in the lifestyle and diet patterns of *H. pylori*-infected individuals [20]. In addition, as discussed below, *H. pylori* infection leads to a reduction in *Lactobacillus*, *Lachnospiraceae*, and *Blautia*, bacteria that have also been found to be reduced in metabolic diseases such as obesity, metabolic syndrome, and diabetes [68,69,70]. Thus, altered intestinal microbiota may lead to disease.

Animal studies showed an interaction between *H. pylori* and normal intestinal microbiota that is associated with intestinal metabolism and inflammation [71]. A study on C57BL/6 mice showed that *H. pylori* infection not only aggravated metabolic disorders induced by an HFD, but also altered the intestinal microbiota. The reported results highlighted that *H. pylori*-infected mice fed with high-fat diets had a different microbiota community, with a significantly higher proportion of Firmicutes and Proteobacteria and a reduction in the population of Bacteroidetes and Verrucomicrobia at the phylum level. In addition, metagenomics analysis showed that *Desulfovibrionaceae* and *Mucispirillum* spp., *Helicobacter*, *Lachnospiraceae* and *Ruminococcaceae* sequences were significantly increased in the *H. pylori*-infected group fed with an HFD [72]. Sex may have a greater impact on gut microbiota composition than diet, as shown by Peng et al. They found that microbiota composition was markedly different between male and female mice independent of diet [73].

Trimethylamine N-oxide is a diet-related microbial metabolite present in seafood and synthesized by hepatic oxidation of trimethylamine, which is produced by enzymatic bacteria in the colon via carnitine and choline. The major source of these compounds in the diet comes from meat, fish, eggs (rich in choline), and dairy products (rich in carnitine) [74,75]. These two compounds are associated with several metabolic and inflammatory disorders [76]. The effect of the presence of trimethylamine N-oxide on inflammation and intestinal microbiota was reported in a study on mice carried out by Wu et al. These authors reported an increase in the expression of growth- and metabolism-associated genes and the urease activity of *H. pylori*, and an increase in the production of virulence factors. The VacA concentration was not increased in the group co-treated with *H. pylori* and trimethylamine N-oxide when compared to the *H. pylori*-treated group, but CagA concentration was significantly higher by treatment with trimethylamine N-oxide when *H. pylori* was co-cultured with GES-1 cells. Thus, trimethylamine N-oxide enhanced *H. pylori* virulence by upregulating the expression of the virulence gene encoding CagA. The authors also confirmed that the intake of trimethylamine enhanced the production of inflammatory markers and reduced the richness and diversity of the intestinal microbiota [77]. Similar results suggested that the treatment of infected mice with choline aggravated both the inflammation produced by the bacteria itself and the alteration of the intestinal microbiota [78]. 

Despite infection by the bacteria altering the profile of the gastric and intestinal microbiota in animals [79,80], it affects the human diurnal oral microbiota [81], causes intestinal dysbiosis in bacterial-induced gastritis [82], and can influence the composition of the gastric microbiota at lower taxonomic levels (in vitro study) [83]. To date, the relationship between microbiota, *H. pylori* infection, and the effect of diet patterns is not fully understood and the number of studies involving the role of diet and its impact on *H. pylori* infection in gastric and intestinal microbiota remains limited (Table 1). Thus, future research will need to decipher the exact mechanisms of *H. pylori* pathogenesis on the *H. pylori*-microbiota axis and extend our understanding of *H. pylori* colonization.

## 3. Management of *Helicobacter pylori* Infection

### 3.1. Current Standard Treatments

There are some guidelines regarding the management of *H. pylori* infection, such as the Maastricht V/Florence Consensus Report [85] and the Taipei global consensus [86]. The latter is specially focused on elucidating the populations that should be screened, and it identified the best treatment (“screen and treat” strategy) to prevent gastric cancer. Although there is no universally accepted regimen for the eradication of *H. pylori*, the current first-line treatment is a triple therapy that combines one proton pump inhibitor and two antibiotics (clarithromycin and amoxicillin) and the quadruple therapy, which includes tetracycline, metronidazole, bismuth salts, and proton-pump inhibitors. However, resistance to clarithromycin has become very prevalent in some geographical areas. In such cases, it can be substituted by metronidazole, although resistance to this antibiotic has already developed in specific areas as well. Aside from drug resistance, antibiotics can also be problematic because they can cause side effects, in part because of the undesired killing of healthy microbiota [87,88,89]. Therefore, owing to a decrease in the therapeutic efficacy of these antibiotic-based therapies, further research is needed in this field. 

### 3.2. New Antibiotics

Some studies have tested new antibiotics or new antibiotic regimes. On one hand, Shi et al. evaluated the efficacy of linezolid and novel oxazolidinone analogs in vitro, by culturing clinical multidrug-resistant *H. pylori* isolates, and in vivo, by orally inoculating mice with these isolates and then administrating the drugs intragastrically. The authors concluded that both, but especially oxazolidinone analogs, were suitable candidates to treat drug-resistant *H. pylori* infections [90]. Linezolid was observed to exhibit in vitro activity, with minimum inhibitory concentrations (MICs) ranging from ≤0.25 mg/L to 32 mg/L against clinical *H. pylori* isolates (MIC_50_, 2 mg/L; MIC_90_, 8 mg/L). The oxazolidinone analogue sy142 inhibited all of the clinical isolates, with MICs of ≤16 mg/L (MIC range, ≤0.25–16 mg/L; MIC_50_, 1 mg/L; MIC_90_, 4 mg/L), even for the multidrug-resistant isolates [90].

On the other hand, Jeong et al. subjected *H. pylori*-infected mice to a gentamicin-intercalated smectite hybrid-based treatment and observed reduced *H. pylori* burden and cytokine secretion, and less atrophy of gastric mucosa, compared to the group treated with standard triple therapy. Importantly, they also analyzed changes in fecal microbiota and concluded that microbiota composition was more similar to uninfected and infected-non-treated groups than microbiota diversity of the standard triple therapy-treated group. These results suggested that this regime may restore gut dysbiosis [91,92]. On the contrary, treatment with antibiotics of *H. pylori*-infected mice fed with a diet high in folate but low in vitamin K caused dysbiosis and hypovitaminosis K, which were likely related to the anemia, gastric hemorrhage, and mortality observed in these animals [93]. 

It is important to remark that there is active research in nanoparticles for targeted antibiotic delivery against *H. pylori* infection [94]. Other therapeutic options, such as nutraceuticals, have also been considered. Finally, there is also an exceptional effort being made on the generation of a prophylactic vaccine against *H. pylori*.

### 3.3. Nutraceutical Approaches

In contrast to antibiotics and pharmacological drugs, nutraceuticals, especially substances found in our daily diet and traditional herbs or plants, have a low probability of toxicity, as they have been consumed for generations in some areas. Consequently, they might be agents for *H. pylori* elimination. Recent literature concerning this subject is summarized in Table 2.

#### 3.3.1. Extracts and Compounds Isolated from Food

*H. pylori* eradication and reversal of gastric mucosa injury was observed in vitro and in vivo using *H. pylori* strains isolated from patients that presented resistance to a variety of antibiotics, and *H. pylori*-infected mice. In this study, the authors confirmed the previously reported antimicrobial activity of ellagic acid, a polyphenol found in walnuts, strawberries, blackberries, raspberries, and pomegranates, among other dietary sources [95]. 

Omega-3 polyunsaturated fatty acids, present in many fish and seafood species, as well as plenty of nuts and seeds, were proposed as a therapy against *H. pylori* infection. A proposed mechanism for this effect is the blocking of the futalosine pathway, an alternative route for menaquinone biosynthesis. Indeed, the blockage of this pathway by supplementation with linoleic acid, an omega-6 fatty acid, inhibited *H. pylori* growth in *H. pylori* cultures and *H. pylori*-infected mice [96]. Some fish and seafood are also rich in other bactericidal compounds. Snakehead fish extract supplementation was described as a useful method to eradicate *H. pylori* in rats, although its superiority when compared to standard first-line antibiotic treatment needs to be confirmed by further studies [97]. By contrast, the effect of the extracts from the *Channa striata* fish on an *H. pylori*-induced gastritis rat model was controversial since it worsened the severity of gastritis when administrated alone, but seemed to contribute to the improvement of the disease when combined with the standard triple therapy [98]. Another study conducted in mice reported that after six weeks of supplementation with astaxanthin from shrimp cephalothorax, *H. pylori*-infected mice showed an increase in splenocytes synthesis of interferon-gamma, IL-2, and IL-10, although the biological meaning of the changes in the cytokine release pattern remains to be explained [99]. 

#### 3.3.2. Traditional Plants or Herbs

There is a vast array of literature aimed at evaluating the potential use of traditional plants or herbs for *H. pylori* eradication or gastroprotection to avoid *H. pylori*-associated health problems. There is a remarkable number of studies focused on folk medicine plants from Asia and Brazil. For example, regarding Asian traditional herbal medicines, in vitro and in vivo results found that Hwanglyeonhaedok-tang (a herb mixture) [100] and yugeunpi (the root bark of *Ulmus davidiana* var. *japonica* (Ulmaceae)) [101] exert anti-inflammatory and anti-*H. pylori* effects. Palmatine, the major component of the traditional Chinese herb *Tinospora sagittata* (Oliv.) *Gagnep.* var. *craveniana* (SY Hu) Lo, improved *H. pylori*-induced chronic atrophic gastritis in rats [103] and eliminated *H. pylori* in infected mice and cell cultures [102]. Patchouli alcohol was described as inhibiting *H. pylori* urease activity and potentiating macrophage digestive activity in vitro [105], as well as limiting the *H. pylori*-promoted recruitment and activation of neutrophils in both rats and culture cells [104]. Berberine was also proposed as a treatment for *H. pylori* given its anti-inflammatory effects in *H. pylori*-induced chronic gastritis in mice [107] and rats [106]. Brazilian folk medicinal plants *Euphorbia umbellata* (leitosinha) [108] and *Piper umbellatum* L. (Piperaceae) [109], among many others, were studied as a potential therapy for *H. pylori*-caused ulcers.

#### 3.3.3. Vitamin D3

Vitamin D3 stimulates innate immune antibacterial activity in several cell types through increase in the production of antimicrobial factors [111]. Accordingly, two different studies concluded that supplementation with vitamin D3 at different doses, for 14 days or 2 months, and treatment with vitamin D3 for 72 h in human normal gastric epithelial cells HFE145 led to *H. pylori* eradication [110,111]. The study conducted by Hu et al. demonstrated that *H. pylori* infection affected Ca^2+^ release from lysosomes of epithelial gastric cells, disrupting lysosomal acidification and allowing the bacteria to remain in the lysosomes. They suggested that vitamin D3 exerted its bactericidal properties by enhancing the lysosomal degradation function via the PDIA3-STAT3-MCOLN3-Ca^2+^ axis [111]. In contrast, Zhou et al. proposed a mechanism wherein vitamin D3 increased the expression of the vitamin D receptor (VDR) and cathelicidin antimicrobial peptides (CAMPs) in the gastric mucosa by binding of the vitamin D3 and VDR complex to the CAMPs promoter [110].

#### 3.3.4. Probiotics

The effect of many lactobacilli species against *H. pylori* was tested in a wide number of publications. While most studies focused on one strain, Lin et al. proposed the combination of *L. fermentum**, L. casei*, and *L. rhamnosus*, given the fact that together, the three probiotics not only eliminated *H. pylori*, but also restored the metabolic imbalance caused by *H. pylori* infection [112]. De Klerk et al. evaluated a panel of lactobacilli species and concluded that *L. gasseri* and *L. brevis* inhibited *H. pylori* attachment to human gastric epithelial cell lines. They also treated transgenic mice expressing human CD46 (to mimic the human stomach) with antibiotics to reduce microbiota abundance. The authors fed animals with *L. gasseri*, and with *H. pylori*, respectively, concluding that *L. gasseri* prevented *H. pylori* infection in this experimental setup [113]. In the same line, *L**. gasseri* improved inflammation associated with *H. pylori* infection in the AGS human gastric adenocarcinoma cell line [114]. Moreover, Chen et al. showed that *L. rhamnosus* and *L. acidophilus* can inhibit the growth of antibiotic-resistant *H. pylori,* its adhesion and invasion to gastric epithelial cells, as well as the production of IL-8 by host cells. Similarly, in mice *L. rhamnosus* and *L. acidophilus* decreased *H. pylori* colonization and *H. pylori*-induced inflammation in the stomach of animals. Remarkably, these probiotics increased the population of beneficial gut bacteria, therefore supporting the reversion of dysbiosis as a mechanism against *H. pylori* infection [115]. Accordingly, treatment with *L. paracasei* decreased *H. pylori* adhesion and *H. pylori*-related inflammation in *H. pylori*-treated AGS human gastric adenocarcinoma cell line and mice. Moreover, histological analysis revealed fewer epithelial lesions in the stomach of *L. paracasei*-fed mice compared to the group that was not treated with the probiotic strain [116]. Furthermore, *L. rhamnosus* isolated from honey or dairy products also eliminated *H. pylori* and improved gastritis caused by this bacterium in mice. The observed effect was comparable to clarithromycin antimicrobial activity, although the use of *L. rhamnosus* seemed to avoid antibiotic-related severe dysbiosis [117,118].

In a mouse model that combined ethanol and *H. pylori* administration, Park et al. treated the animals with green tea-derived *L. plantarum*, concluding that these probiotics decreased inflammation and improved gastric ulcers, possibly by restoring healthy gut microbiota [119]. In another study by the same group, mice were fed *L. plantarum* before *H. pylori* infection. They reported that *H. pylori*-induced gastric mucosa inflammation and gastric dysbiosis could be prevented by pre-treatment with *L. plantarum* [120].

Finally, dead *L. johnsonii* was reported to eradicate *H. pylori* in vitro and from the stomach of germ-free mice, although the molecular mechanism was not clear [121].

To conclude, we mainly found studies reporting the effects of lactobacilli species against *H. pylori* infection. To date, it is strongly believed that probiotics can be useful for *H. pylori* eradication, although still with controversial results [87,122]. Nevertheless, more studies aimed at deciphering molecular mechanisms may shed light on this question.

## 4. Discussion

*H. pylori* infection is the major cause of the development of gastric and duodenal ulcers and gastric cancer worldwide, affecting almost 50% of the world’s population. Infected individuals can incur direct damage in the epithelial cells, or damage induced by the pro-inflammatory factors stimulated after infection. The affectation depends on the host’s genetic predisposition as well as other conditions (e.g., diet, age, or an altered intestinal microbiota).

Bearing this in mind, in the present review, we described the impact of dietary patterns on *H. pylori* infection and the modulation of microbiota to counteract its effect.

We highlighted studies that reported alterations in the microbiota caused by the presence of this bacterium and their relation to diet. Our search was focused mainly on animal studies because they present several advantages, including their reproducibility, reliability, affordability, and technical availability when compared with human studies. Our purpose was also to identify relevant recent studies addressed to treat *H. pylori* infection using extracts and compounds isolated from food and traditional plants or herbs reported in animal models covering the period of 2016–2020. 

Gut microbiota itself, as well as dietary patterns such as high salt or sugar intake, seem to have a notable impact on the development, proliferation, and increased risk of developing more severe conditions after *H. pylori* infection. 

By contrast, foods such as whole-grains, roots and tubers, vegetables, mushrooms, miscellaneous beans, vegetable oils, nuts or seeds consumed in abundance are linked to a reduced risk of infection. Of note, the majority of these mentioned foods have proven prebiotic, probiotic, and anti-inflammatory properties which, in turn, favor a healthy microbiota. Thus, accounting for the fact that diet has a considerable effect on the composition of the gastrointestinal microbiota and that *H. pylori* colonization results in alterations of gastric microbiota and reduction in bacterial diversity, it seems evident that there are mutual interactions between all three factors. However, the relationship that each of these variables has with one another remains poorly understood. Consequently, more studies connecting the gut with microbiota-host-*H. pylori* and dietary patterns interactions are required to understand the complete scenario of these associations and their effect on gastrointestinal health. 

Studies conducted on humans have focused on the analysis of the diversity and variations in the microbiota of *H. pylori* infected and uninfected individuals [123], and none have considered the effect of diet. 

Current standard treatments include antibiotics therapy, although this approach presents two major inconveniences: (1) the undesired killing of a healthy microbiota, and (2) the increased resistance of *H. pylori* to commonly used antibiotics. 

In addition, the emergence of strains of *H. pylori* that are increasingly resistant to the use of antibiotics and the poor adherence to treatment by patients means that the efficacy of the use of these therapies is highly compromised, sometimes reaching ≤80% [124]. Alternative therapies involving the use of natural substances or compounds of certain foods are currently being pursued. Many studies have confirmed the antibacterial activity of substances in food, although the number of prospective randomized trials assessing their use in clinical practice remains scarce. Moreover, some diets contain many substances with potent antibacterial activity against *H. pylori* and dietary interventions may enable a decrease in *H. pylori* colonization and lead to the reduction of gastritis and gastrointestinal-associated diseases [24]. 

On the other hand, infection with *H. pylori* is more prevalent in developing countries [4]. Treatment regimens based on antibiotics or other drugs have a lower chance of success in these countries because socioeconomic issues limit their accessibility and availability. Thus, the establishment of alternate approaches, such as nutritional intervention, seem a suitable tool for reducing *H. pylori* infection and its undesired effects on gastrointestinal health. Dietary interventions and the use of supplements, such as probiotics or prebiotics, seems a more affordable strategy to, at the very least, ameliorate the damage induced by *H. pylori* in infected individuals. Thus, policies in this regard should be made available to the entire population.

In conclusion: (1) *H. pylori* infection is still an unsolved problem with potentially severe complications for gastrointestinal health, (2) recent results from studies, mainly conducted on animals models, show evidence of the mutual interactions between, diet, gastrointestinal microbiota, and *H. pylori*, and (3) more research aimed at shedding light on the molecular mechanisms involving diet, gastrointestinal microbiota, and *H. pylori* is needed and may prove helpful in the development of novel therapeutic approaches.

## Figures and Tables

**Figure 1 pathogens-10-00875-f001:**
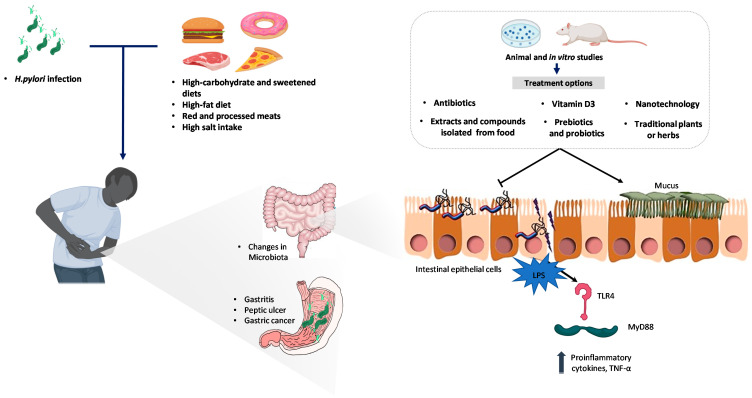
*H. pylori* invades the human stomach, negatively impacting host health, which is promoted or aggravated by specific dietary patterns. The main effects are gastritis, which can lead to peptic ulcers and eventually gastric cancer, alterations in the gut microbiota, and inflammation. Animal and in vitro models are routinely used to test the efficacy of emerging treatments for the eradication of *H. pylori*. These approaches mainly focus on the inhibition or reversal of adverse effects by avoiding the attachment and colonization of host intestinal epithelial cells and the attenuation of the consequences of the host´s pro-inflammatory state after *H. pylori* infection. Abbreviations: LPS, lipopolysaccharide, TLR4, toll-like receptor 4, and TNF-α, tumor necrosis factor-alpha.

**Table 1 pathogens-10-00875-t001:** Summary of recent studies linking the effect of diet on *H. pylori* and microbiota.

Dietary Pattern	Animal Model (*n*)	Effects	Main Effects in Microbiota	Reference
(1) Excessive salt intake (2) High-salt and low-iron diet	(1) Mongolian gerbil *n* =108 (2) Mongolian gerbil model *n* = 96	↑ Colonization by *H. pylori* (1) ↑ Risk of gastric cancer, hypochlorhydria and epithelial damage, Loss of parietal cells, and intestinal metaplasia (2) ↓ Hemoglobin levels	NA	[18,27,33]
Cured, pickled, and smoked foods	Rhesus monkey *n*= 23	Induction of intestinal metaplasia and intraepithelial neoplasia ↑ Expression of oncogenes	NA	[46]
Diet poor in iron	Mongolian gerbil *n* = 10	↑ Infiltration of immune cells at the site of infection ↑ Onset of gastritis ↑ Rate of cancer development	NA	[41]
High-fat diet	C57BL/6 mice *n* = 10	Intestinal microbiota alteration	↑ Firmicutes and Proteobacteria ↓ Bacteroidetes and Verrucomicrobia	[72]
High-fat diet	Specific-pathogen-free C57BL/6 mice *n* = 10	*H. pylori* infection and high-fat diet promote dysbiosis intestinal microbiota	↑ Firmicutes/Bacteroidetes (F/B) ratio ↑ *Prevotellaceae*-UCG-001, *Helicobacter*, and *Rikenella* ↓ *Blautia*, *Lactobacillus*, and *Lachnoclostridium*	[84]
Meat, fish, eggs, and dairy products	BALB/c female mice *n* = 40	TMA and TMAO taken by diet induced: ↑ *H. pylori* development ↓ Intestinal microbiota richness and diversity	↓ *Anaerovorax*, *Rikenellaceae* RC9 gut group, *Lachnospraceae* UCG 008 and *Parabacteroides* ↑ *Escherichia*/*Shigella*	[77]

Abbreviations: *n*, number of animals; NA, not analyzed; TMA, trimethylamine; and TMAO, Trimethylamine N-oxide.

**Table 2 pathogens-10-00875-t002:** Nutraceutical approaches to treat *H. pylori* infection reported in animal models (2016–2020).

Nutraceutical Tested	Type and Nature of Strain Analyzed	Sensitivity Profile	Animal Model (*n*)	Effects	Main Effects in Microbiota	References
Extracts and compounds isolated from food						
Ellagic acid	Sydney Strain 1 isolated from gastroduodenal patients	In vitro minimum inhibitory concentration 15 mg/L ellagic acid, 0.015 mg/L clarithromycin, 0.5 mg/L amoxicillin, 0.25 mg/L metronidazole (resistant)	C57BL/6 mice *n* = 6	Elimination of *H. pylori* in cultures and mice Reversion on *H. pylori*-induced gastric mucosa injury in mice	NA	[95]
Linoleic acid	The nature of the strain: Not indicated. Sydney Strain 1 and TN2GF4 strains	Not indicated	C57BL/6NCrl mice *n* = 6	Inhibition of *H. pylori* growth	NA	[96]
Snakehead fish extract	The nature of the strain: Gastric biopsy specimens of duodenal ulcer patients. Strain: unknown	Not indicated	Albino rats *n* = 7	Elimination of *H. pylori*	NA	[97]
*Channa striata* fish extract	The nature of the strain: Gastric biopsy specimens of duodenal ulcer patients. Strain: unknown	Not indicated	Albino rats *n* = 7	In combination with standard triple therapy, reversion of gastritis	NA	[98]
Astaxanthin from shrimp cephalothorax	The nature of the strain: Gastric biopsy specimens of duodenal ulcer patients. J99 strain	Not indicated	BALB/c mice *n* = 40	↑synthesis of IFN-γ, IL-2 and IL-10 in splenocytes in infected mice	NA	[99]
Traditional plants or herbs						
Hwanglyeonhaedok-tang	The nature of the strain: Not indicated. Sydney Strain 1	In vitro minimum inhibitory concentration: 400 to 1600 μg/mL Hwanglyeonhaedok-tang; 0.00396 ~ 0.125 µg/mL amoxicillin; 0.001953 ~ 32 µg/mL clarithromycin	C57BL/6 mice *n* = 7	Elimination of *H. pylori* cultures and mice ↓ *H. pylori*-induced inflammation in cultures and mice	NA	[100]
Yugeunpi	The nature of the strain: Not indicated. 51 and 43,504 strains	In vitro minimum inhibitory concentration: 25 and 50 µM bioactive compounds of Yugeunpi: (2R,3S)-2-ethoxychroman-3,5,7-triol-7-O-β-d-apiofuranoside, fraxetin, 4-O-β-d-glucopyranosyl vanillic acid, syringic acid	In vitro Murine microglia BV-2 cell line	Elimination of *H. pylori* in cultures ↓ *H. pylori*-induced inflammation in cultures	NA	[101]
Palmatine	The nature of the strain: (1): not indicated; (2): Chronic atrophic gastritis. (1): Sydney Strain 1; (2): SCYA201401 and Sydney Strain 1 strains	(2) In vitro minimum inhibitory concentration: strain SCYA201401 6.25 μg /mL Palmitine; Strain Sydney Strain 1, 3.12 μg/mL Palmitine	(1) Male Sprague-Dawley rats *n* = 6, (2) C57BL/6 mice *n* = 8	↓ *H. pylori*-induced chronic atrophic gastritis in rats Elimination of *H. pylori* in cultures and mice	NA	[102,103]
Patchouli alcohol	The nature of the strain: (1): not indicated; (2): not indicated. (1): NCTC11637; (2): ATCC43504	Not indicated	Male Sprague-Dawley rats *n* = 6	(1) Inhibition of *H. pylori* urease activity (2) ↑ macrophage digestive activity in cultures ↓ *H. pylori*-promoted recruitment and activation of neutrophils in cultures and rats	NA	[104,105]
Berberine	The nature of the strain: (1): not indicated; (2): stomach tissue of a male chronic atrophic gastritis patient No. ZCDC111001. (1): CagA+/VacA+ *H. pylori* strain 342	Not indicated	(1) C57Bl/6 mice, *n* = not indicated (2) Sprague-Dawley rats *n* = 6	Attenuates *H. pylori-*induced inflammation and chronic gastritis in mice and rats	NA	[106,107]
*Euphorbia umbellata*	The nature of the strain: not indicated	Not indicated	Wistar rats *n* = 7	Heals *H. pylori*-caused ulcers in rats	NA	[108]
*Piper umbellatum* L.	The nature of the strain: not indicated. ATCC 43,504 (vacA and cagA positives) strain	Not indicated	Swiss-Webster mice, *n* = 8	Heals *H. pylori*-caused ulcers in mice	NA	[109]
Vitamin D3	The nature of the strain: (1), (2): not indicated. (1) Sydney Strain 1 (2) Sydney Strain (SS)1 (VacA+ and CagA+) strain	Not indicated	(1) C57BL/6 mice *n* = 8 (2) C57BL/6 mice *n* = 5	Elimination of *H. pylori*	NA	[110,111]
Probiotics						
*L. fermentum, L. casei* and *L. rhamnosus*	The nature of the strain: gastritis patient. 26,695 (ATCC 700392) strain	Not indicated	C57BL/6 mice *n* = 10	Elimination of *H. pylori* Restoration of *H. pylori*-induced metabolic imbalance	NA	[112]
*L. gasseri* and *L. brevis*	The nature of the strain: not indicated. Sydney Strain 1	Not indicated	hCD46Ge transgenic mouse line (CD46+/+) *n* = 6	Inhibit *H. pylori* attachment in cultures Prevent *H. pylori* infection in mice	NA	[113]
*L. gasseri*	The nature of the strain: isolated from individual infected patients with chronic gastritis and peptic ulcer disease. Strains: OC168, OC235, OC250, OC824, OC912, OC562, OC576, OC722, OC803, OC805	Not indicated	In vitro. Human gastric adenocarcinoma epithelial AGS cell line	↓*H. pylori*-induced inflammation	NA	[114]
*L. rhamnosus* and *L. acidophilus*	The nature of the strain: not indicated. 26695 (ATCC 700392) strain	Not indicated	BALB/c mice *n* = 4–5	Inhibition of growth, adhesion and invasion of *H. pylori* in cultures ↓ *H. pylori*-induced IL-8 production in cultures ↓ *H. pylori* colonization and *H. pylori*-induced inflammation in mice	*L. rhamnosus* associated ↑ *Bifidobacterium*, Proteobacteria, and *A. muciniphila*. *L. acidophilus* associated ↓ Actinobacteria and *E. coli*, ↑ Proteobacteria and *A. muciniphila*	[115]
*L. paracasei*	The nature of the strain: not indicated. Sydney Strain 1 strain	Not indicated	C57BL/6 mice *n* = 10	↓*H. pylori* adhesion and *H. pylori*-related inflammation (IL-8 expression) in cultures and mice. ↓ epithelial lesions in the stomach of mice	NA	[116]
*L. rhamnosus*	The nature of the strain: not indicated. (1) ATCC43504 strain. (2) 26695 ATCC 700392 strain	Not indicated	(1) C57BL/6 mice *n* = 8, (2) C57BL/6 mice *n* = 10	Elimination of *H. pylori* ↓*H. pylori*-caused gastritis	The treatment maintains the balance lactic acid bacteria/coliform bacteria	[117,118]
*L. plantarum*	The nature of the strain: not indicated. (1) Sydney Strain 1 strain (2) Sydney Strain 1 (HpKTCC) strain	Not indicated	(1) (2) C57BL/6J mice *n* = 8 *H. pylori* and ethanol treatment	↓inflammation ↓gastric ulcers in mice treated with *H. pylori* and ethanol Prevents *H. pylori*-induced gastric mucosa inflammation and gastric dysbiosis in mice	↑ *Bifidobacterium* spp. and *Clostridium butyricum*. Prevention of the decrease in Shannon’s diversity index and Simpson’s diversity index	[119,120]
Dead *L. johnsonii*	The nature of the strain: not indicated. *H. pylori* No.130 strain	Not indicated	Germ-free Balb/c mice *n* = 10	Elimination of *H. pylori* in cultures and mice	NA	[121]

Abbreviations: *n*, number of animals; NA, not analyzed; IFN-γ, interferon gamma; IL-2, interleukin-2; IL-10, interleukin-10; and IL-8, interleukin-8.

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
