# Peer review of "Impact of Dietary Patterns on H. pylori Infection and the Modulation of Microbiota to Counteract Its Effect. A Narrative Review"

_pathogens, 2021, doi:10.3390/pathogens10070875_

Round 1

Reviewer 1 Report

This manuscript was well-revised compared to the previous version.

I would recommend modifying the presentation of the new information added to table 2, as in a present way, the first column is way too hard to clearly understand. Please consider adding one or two supplementary columns.

Author Response

Comment #1

This manuscript was well-revised compared to the previous version. I would recommend modifying the presentation of the new information added to table 2, as in a present way, the first column is way too hard to clearly understand. Please consider adding one or two supplementary columns.

Response: The authors would like to thank the reviewer for his respected comments and effort made during the review process, which are highly appreciated. Table 2 was modified according to the reviewer’s comments.

Reviewer 2 Report

The review is almost interesting and well written.

I have only few suggestions:

  • the insertion of a graphic abstract to summarize the entire review
  • add a separate section for the microbiome gut as it is revealing in the title and
  • add about the gut microbiome ref. DOI: 10.3390/biom9060237

Author Response

Comment #1

The review is almost interesting and well written. I have only few suggestions: The insertion of a graphic abstract to summarize the entire review add a separate section for the microbiome gut as it is revealing in the title and add about the gut microbiome ref. DOI: 10.3390/biom9060237

Response: Thank you to the reviewer for his/her comment, we add new information about gut microbiome in the section 2.1. of our review and also we have created a graphical abstract, the main text now state (page 5, lines 173-182).

Lines 173-182

The gut microbiome is the genetic material of all the microbes that live on and inside the human body [53-55]. H. pylori colonization results in alterations of gastric microbiota and reduction in bacterial diversity [56,57] because could produce alterations in the host by modifying the gut microbiome [58]. A growing number of studies are reporting greater ecosystem diversity in the gastrointestinal tract and associating the H. pylori presence with variations within the structure of the microbiome [59]. Gut microbiome changes triggered by the first H. pylori acquisition could also corroborate the host immunological status and, as a result, the materialize of several diseases. This is important due to the gut microbiota is modified by antibiotics treatment, H. pylori-induced inflammation is decreased [60].
